# Moderate–Vigorous Physical Activity and Clinical Outcomes in Adults with Nondialysis Chronic Kidney Disease

**DOI:** 10.3390/jcm10153365

**Published:** 2021-07-29

**Authors:** Ji Hye Kim, Young Youl Hyun, Kyu-Beck Lee, Sung Woo Lee, Hayne Cho Park, Wookyung Chung, Joongyub Lee, Yun Kyu Oh, Kook-Hwan Oh, Dong-Wan Chae, Curie Ahn

**Affiliations:** 1Division of Nephrology, Department of Internal Medicine, Kangbuk Samsung Hospital, Sungkyunkwan University School of Medicine, 29 Saemunan-ro, Jongno-gu, Seoul 03181, Korea; jihyek24@gmail.com (J.H.K.); yy.hyun@samsung.com (Y.Y.H.); 2Eulji Medical Center, Department of Internal Medicine, Eulji University, Seoul 01830, Korea; neplsw@eulji.ac.kr; 3Department of Internal Medicine, Kangnam Sacred Heart Hospital, Hallym University Medical Center, Seoul 07742, Korea; hanepark798@gmail.com; 4Department of Internal Medicine, Gil Hospital, Gachon University, Incheon 21565, Korea; jwkpsj79@hamail.net; 5Department of Prevention and Management, Inha University Hospital, Inha University School of Medicine, Incheon 22332, Korea; denver261@gmail.com; 6Department of Internal Medicine, Seoul National University Boramae Hospital, Seoul 07061, Korea; yoonkyuoh@gmail.com; 7Department of Internal Medicine, Seoul National University Hospital, Seoul National University College of Medicine, Seoul 03080, Korea; ohchris@hanmail.net; 8Department of Internal Medicine, Seoul National University Bundang Hospital, Seongnam-si 13620, Korea; cdw1302@snubh.org; 9Department of Internal Medicine, National Medical Center, Seoul 04564, Korea; curieoffice@gmail.com

**Keywords:** physical activity, clinical outcomes, chronic kidney disease

## Abstract

The health benefits of physical activity (PA) are well known. However, the association between an adequate amount of moderate–vigorous PA (MVPA) and clinical outcomes has limited evidence in chronic kidney disease (CKD). We assessed PA using a self-administered questionnaire. The amount of MVPA was categorized into four groups: none, low, moderate, and high (0, <7.5, 7.5–14.9, and 15.0–29.9 metabolic equivalent-hours/week, respectively). We analyzed the association between the amount of MVPA and clinical outcomes. Among a total of 1909 adults with CKD, adults with MVPA showed various beneficial outcomes compared to those with no MVPA in a Kaplan–Meier curve followed over a median of 5.9 years. In multivariable-adjusted Cox proportional hazard models, a low and a moderate amount of MVPA was associated with a lower risk of all-cause death. A moderate amount of MVPA was associated with a lower risk of cardiovascular events. A high amount of MVPA was associated with a lower risk of end-stage kidney disease in ESKD in 1324 adults with eGFR <60 mL/min/1.73 m^2^. Age and sex modified the relationships between MVPA and clinical outcomes. MVPA is associated with various beneficial outcomes across the amount of MVPA. PA plans should be tailored for individual adults with CKD.

## 1. Introduction

The health benefits of physical activity (PA) are well established. It is associated with primary prevention and better clinical outcomes in chronic diseases, including cardiovascular disease (CVD), diabetes, hypertension, cancer, depression, and osteoporosis [1,2]. PA of moderate- and vigorous-intensity aerobic activity has been successfully categorized by questionnaires [3] and is evidenced to improve cardiovascular function [4,5] and health outcomes [6,7]. The 2018 PA for Americans guideline [8] suggests that active adults achieve at least 150 min/week of moderate-intensity PA, 75 min/week of vigorous-intensity PA, or an equivalent combination of moderate–vigorous PA (MVPA). The 2020 World Health Organization (WHO) guideline [9] recommends that adults should perform 150–300 min/week of moderate-intensity PA, 75–150 min/week of vigorous-intensity PA, or an equivalent combination of MVPA. However, studies of the benefits and harms associated with amounts of MVPA have provided limited evidence among adults with chronic kidney disease (CKD).

CKD is a public health problem that amplifies the risks of mortality, cardiovascular events (CVEs), and progression to end-stage kidney disease (ESKD). Adults with CKD have reduced physical function and are not as aerobically fit as the general population. A meta-analysis of clinical trials of regular exercise training in adults with CKD demonstrated that regular exercise training improved aerobic capacity, muscle function, cardiovascular function, and health-related quality of life [10]. A recent systematic review of cohort studies reported that survival rates correlated with greater PA in nondialysis CKD [11]. The lifestyle section of the 2012 Kidney Disease Improving Global Outcomes Guideline (KDIGO) for CKD [12] recommends that individuals with CKD engage in PA for at least 30 min/day and 5 times/week. PA might have a dose–response curve with a plateau in health benefits [13]. There are four types of PA: aerobic, resistant, balance, and flexibility, comprising frequency, intensity, and time [14,15]. Despite the encouragement of PA in managing CKD, there is limited evidence about the relationship of the amount of PA with clinical outcomes in adults with nondialysis CKD.

Although MVPA is beneficial for various chronic diseases, adults with CKD have fatigue, weakness, reduced exercise capacity, and multiple comorbidities, which make it difficult to engage in large amounts of MVPA [16]. Moreover, vigorous-intensity PA could increase the risk of musculoskeletal and cardiovascular complications in CKD patients with multiple comorbidities [17,18]. The objectives of this prospective CKD cohort study were to (1) evaluate the patient characteristics that contribute to the amount of MVPA; (2) investigate the association between different amounts of MVPA and the risks of all-cause death, CVE, and ESKD, and (3) evaluate the association in subgroups according to age and sex.

## 2. Method

### 2.1. Study Design and Participants

The KoreaN cohort study for Outcome in patients With CKD (KNOW-CKD) is a nationwide, multicenter prospective study investigating the clinical outcomes of Koreans with non-dialysis-dependent CKD. The design of the KNOW-CKD study has been published (NCT01630486 at http://www.clinicaltrials.gov, accessed on 5 June 2019) [19]. Nine clinical centers in university-affiliated hospitals enrolled 2238 adults between 20 and 75 years of age with CKD at all stages from 2011 to 2016. Subjects were excluded if they had a history of malignancy, advanced heart failure, a single kidney, liver cirrhosis, chronic lung disease, or other factors according to the study protocol. We analyzed 1909 participants from this cohort who underwent extensive laboratory tests and completed the health questionnaire and PA questionnaire (Appendix A). The principles of the Declaration of Helsinki were followed. The study protocols were approved by the Institutional Review Board (1104-089-359), and informed consent was obtained.

### 2.2. Data Collection and Measurement

Information about patient demographic characteristics, medical history, and lifestyle factors was collected by self-report and a review of the medical records. Baseline demographics and health questionnaires [20] were retrieved from the electronic data management system (PhactaX) of Seoul National University Medical Research Collaborating Center. We assessed current smoking, alcohol consumption, and comorbidities that were defined based on the study protocol. Serum creatinine was measured using an isotope dilution mass spectrometry-calibrated method at a central laboratory. The estimated glomerular filtration rate (eGFR) was calculated by using the CKD Epidemiology Collaboration equation. Cardiovascular events (CVEs) were defined as the first occurrence of a nonfatal CVE during follow up. Nonfatal CVE included any nonfatal coronary artery events (unstable angina, myocardial infarction, coronary intervention, or coronary surgery), hospitalization for heart failure, ischemic or hemorrhagic stroke, or symptomatic arrhythmia. ESKD was defined as reduced renal function requiring dialysis or kidney transplantation.

### 2.3. Assessment of Physical Activity

Caspersen et al. and WHO define PA as any bodily movement produced by skeletal muscles that requires energy expenditure including activities performed while working, playing, carrying out household chores, traveling, or engaging in recreational pursuits [21]. Aerobic PA is defined as any activity that uses large muscle groups and can be maintained continuously and rhythmically. In this study, PA type, intensity, and amount were measured using the Korean form of the International Physical Activity Questionnaire (K-IPAQ). The K-IPAQ is a self-administered, validated questionnaire composed of 8 items with illustrations (Appendix A) [22,23]. Four of the questions asked specifically about the frequency, intensity, and duration of MVPA during work and leisure activities. The specific PA performed at various intensities is represented by metabolic equivalent (MET) levels [24]. One MET represents the energy expenditure of a resting person. In the K-IPAQ, PA intensity was classified as moderate PA and vigorous PA. Each intensity of PA was assigned a corresponding MET; moderate intensity—3 MET and vigorous intensity—6 MET. Bouts of MVPA lasting less than 10 min were not counted, and those greater than 3 h were truncated. The PA guidelines recommend the weekly amount of 150 min moderate-intensity PA or 75 min vigorous-intensity PA or equivalent is the same as 7.5 MET-hour/week. We calculated the amount of MVPA (MET-hours) as the products of intensity (MET) and weekly MVPA time (hours). We then categorized the amount of MVPA into 4 groups; none (0 MET-hour/week), low (<7.5 MET-hour/week), moderate (7.5–14.9 MET-hour/week), and high (15.0–29.9 MET-hours/week) in accordance with previous studies [25,26].

### 2.4. Covariates

We included several covariates in our modeling of the association between MVPA and clinical outcomes: demographic, lifestyle factors, and clinical data. The demographic variables were age, sex, education (high-school graduate or below, college graduate or above), and work (employed, unemployed). The lifestyle factors were smoking status (current or not) and alcohol consumption (more than over 2 times of 2 standard drinks/week or not). The clinical data were eGFR, urine albumin–creatinine ratio (ACR), systolic blood pressure, serum albumin, and history of diabetes mellitus or CVD. Employed individuals were those performing paid work for more than 1 h/week or support work for more than 18 h/week according to the International Labor Organization and the Korean Ministry of Employment and Labor.

### 2.5. Statistical Analyses

Data are expressed as the mean ± standard deviation or median (interquartile range), as appropriate. Baseline characteristics were summarized descriptively according to the amount of MVPA. Linear trends in categories were tested using linear regression analysis.

We analyzed the factors associated with the amount of MVPA. Multivariable logistic regression was used to determine the association between MVPA and demographic, lifestyle, and clinical factors. We longitudinally analyzed the association between the amount of MVPA and adverse clinical outcomes. Kaplan–Meier curves were generated to assess all-cause death, CVE, and ESKD according to the 4 levels of amounts of MVPA. Cox proportional hazards models were used to determine the association between MVPA and adverse clinical outcomes after adjusting for the demographic, clinical, and lifestyle factors. Model 1 was adjusted for age, sex, eGFR, and the natural log of albuminuria. Model 2 was adjusted for the Model 1 variables in addition to education, work status, smoking, alcohol consumption, diabetes, CVD, body mass index, systolic blood pressure, and serum albumin. Model 3 was adjusted for Model 2 variables plus statin use and angiotensin-converting enzyme inhibitor or angiotensin receptor blocker use. Harrell’s C-statistics were used to evaluate the goodness of fit of the models. The concordance index (Harrell’s C-index) quantifies a model’s predictive discrimination ability. The C-index is the probability of concordance between the predicted and observed outcomes, with values ranging from 0.5 (no discrimination) to 1.0 (perfect discrimination). All statistical analyses were performed using Stata Version 16 (StataCorp, College Station, TX, USA).

## 3. Results

### 3.1. Description of the Population

The baseline characteristics of the study population are presented according to the amount of MVPA (Table 1). At baseline, the average age of participants was 53 ± 12 years, and 62% were male. Among the 1909 study subjects, 964 (50.5%) performed no MVPA, and 509 (26.7%) performed MVPA that met the American and WHO guidelines (>7.5 MET-hours/week). The more active participants tended to be younger, male, highly educated, employed, more likely to drink alcohol, and less likely to have CVD and diabetes compared to those with no MVPA.

### 3.2. Factors Associated with MVPA

The factors associated with MVPA are presented in Table 2. In the univariable logistic regression analysis, MVPA was associated with male sex, age of 40–49 years, nondiabetes, non-CVD, eGFR of 60–89 mg/min/1.73 m^2^ (CKD stage 2), high education, and employment. We found that participants in their 40 s had the highest amount of MVPA, and those with CKD stage 2 had the highest amount of MVPA by eGFR category. The multivariable logistic regression analysis showed that MVPA was associated with male sex, age of 40–49 years, nondiabetes, and CKD stage 2. Participants in the 60–75 years age group performed less MVPA than those in their 40s (OR = 0.71; 95% CI: 0.53–0.95). Participants with advanced CKD (eGFR < 30 mL/min/1.73 m^2^) performed less MVPA than those with CKD stage 2 (OR = 0.69, 95% CI: 0.52–0.92). The factors associated with MVPA over 7.5 MET-hour/week showed patterns similar to the factors associated with any MVPA.

### 3.3. Association between the Amount of MVPA and Clinical Outcomes

The Kaplan–Meier curves showed the time to all-cause death, CVE, and ESKD according to the amount categories of MVPA. All-cause death had fewer events in participants with any amount of MVPA compared to those with no MVPA (Figure 1). CVE had fewer events in participants with a moderate amount of MVPA compared to those with no MVPA (Figure 2). ESKD had fewer events in participants with a high amount of MVPA compared to those with no MVPA (Figure 3).

The associations between the amount of MVPA and clinical outcomes are shown in Table 3. The risk of all-cause death was lower in adults with low and moderate MVPA than in those with no MVPA in both the crude and adjusted models. The hazard ratios (HRs) for death in adults with low, moderate, and high MVPA were 0.44 (95% CI: 0.25–0.77), 0.42 (95% CI: 0.20–0.87), and 0.88 (95% CI: 0.45–1.74), respectively, in Model 2 (C statistic index = 0.844). The CVE risk was lower in adults with moderate MVPA than in those with no MVPA in both the crude and adjusted models. The HRs for CVE in adults with low, moderate, and high MVPA were 1.00 (95% CI: 0.67–1.51), 0.47 (95% CI: 0.24–0.89), and 0.89 (95% CI: 0.49–1.62), respectively, in Model 2 (C statistic index = 0.771). The incident ESKD was lower in high MVPA group than in the no MVPA group in the crude model [HR: 0.53 (95% CI: 0.38–0.75)] and in Model 1 [HR: 0.65 (95% CI: 0.45–0.95)], but the association was lost in Model 2 [HR: 0.72 (95% CI: 0.49–1.06), *p* = 0.091] (C statistic index = 0.911).

MVPA predicted the risk of all-cause death (C statistic index = 0.609) better than it predicted the risk of CVE and ESKD based on Harrell’s C-statistics (C statistic index = 0.554, and 0.547, respectively, in the crude model). MVPA had a closer association with the risk of all-cause death than the risk of CVE and ESKD.

### 3.4. Subgroups

When subgroups on the basis of age, sex, and eGFR stage were conducted, we observed modifications of the association between the amount of MVPA and clinical outcomes in nondialysis CKD patients (Figure 4). The association between the amount of MVPA and the risk of all-cause death was more prominent among age <60 years than among age ≥60 years (*p* for interaction = 0.014) and among females than among males (*p* for interaction = 0.005). The association between the amount of MVPA and the risk of CVE was not different according to age and gender groups. The association between MVPA and mortality was more prominent among younger and females. Thus, age and sex modified the relationship between MVPA and mortality.

In a total of 1909 adults with CKD, the HRs for all-cause death and CVE in adults with MVPA were 0.51 (95% CI: 0.33–0.77) and 0.83 (95% CI: 0.59–1.15). The risk reduction of all-cause death was more prominent than that of CVE. The risk reduction of ESKD was not significant in a total of 1909 adults with CKD, but the high amount of MVPA was associated with reduced risk of ESKD in 1324 adults with eGFR < 60 mL/min/1.73 m^2^ [HR: 0.53, 95% CI: 0.35–0.82].

## 4. Discussion

In the KNOW-CKD study, the factors associated with the amount of MVPA were sex, age, diabetes, and eGFR stage. The lower risk of all-cause death and CVE occurred in adults with a moderate amount of MVPA. Moreover, even a low amount of MVPA was associated with a lower risk of death, and a high amount of MVPA was associated with a lower risk of ESKD. The risk of clinical outcome varied in subgroups divided by age and sex. Our study suggests that the beneficial effects of MVPA differ with type of outcome, patient’s age, sex. Therefore, planning individualized MVPA would be better for adults with CKD.

PA and exercise are principal interventions in the prevention and treatment of chronic diseases, including CVD, type 2 diabetes, cancer, lung disease, osteoporosis, and neuropsychiatric diseases [2]. PA is a multifaceted behavior composed of individual frequency, intensity, time, and type (FITT) [14]. The FITT principle is critical in exercise prescriptions [15], but few studies have considered how FITT influences the outcomes of specific chronic diseases. Resistant PA mainly strengthens muscle and bone. Aerobic PA, in which the muscles move in a rhythmic manner for a sustained period, relies on aerobic metabolism to extract energy and improve cardiovascular and respiratory function. Current PA guidelines [8,9] imply that MVPA of at least moderate-intensity for 150 min/week or vigorous-intensity for 75 min/week (7.5 MET-hour/week) [25,26], regardless of whether it is performed at a moderate or vigorous intensity, is key for health benefits. The lifestyle section of the 2012 KDIGO for CKD [12] recommends that individuals with CKD engage in PA for at least 30 min/day, 5 times/week (evidence level: 1D). The lifestyle section of the 2020 KDIGO for diabetes and CKD [27] recommends that adults with diabetes and CKD perform moderate-intensity PA for at least 150 min/week or to a level compatible with their cardiovascular and physical tolerance (evidence level: 1D). There is wide variability in PA patterns and response of cardiorespiratory fitness among individuals with chronic disease. The question thus remains: what dose of PA will provide health benefits in CKD?

Adults with CKD have weakness, loss of muscle mass, and poor physical performance. A large amount of MVPA could increase the risk of musculoskeletal and cardiovascular complications in CKD. A recent review described that quantifying the lowest and the highest levels of exercise that confer health benefits is important in exercise prescriptions [28]. Overall, 10–40 MET-hours/week of total PA appears to be the most efficient prescription (sweet spot) for adults with CVD, but limited evidence is available about the appropriate amount of MVPA for adults with CKD. Previous studies showed beneficial health effects from exercise training in CKD. A meta-analysis of 41 clinical trials found that exercise training improved aerobic capacity, muscular function, cardiovascular function, and quality of life [10]. Another meta-analysis of 59 clinical trials reported that the benefits of exercise are well established and that the evidence is strong for the effects of aerobic exercise in improving physical fitness, muscular strength, and quality of life in dialysis patients [29]. A recent systematic review of observational studies reported that survival rates correlated with greater PA in nondialysis CKD [11]. However, most of these studies included small samples and short-time follow up and applied exercise to patients on maintenance hemodialysis. Moreover, these studies had large heterogeneities in exercise types, measurement, and outcomes. Laboratory-based measurements, such as VO2 measurement, accelerometer, or isokinetic dynamometer, are complex and impractical in clinical applications [3]. As a pragmatic measurement of MVPA, the IPAQ can be quickly and easily conducted and has been used to determine worldwide PA guidelines [24]. All-cause death, CVE, and ESKD are essential clinical outcomes in nondialysis CKD. Therefore, we analyzed the prospective MVPA data using the K-IPAQ and the hard clinical end-points in nondialysis CKD patients.

An observational study of the National Health and Nutrition Examination Survey (NHANES) data of 15,368 adults with CKD showed that physical inactivity was more common among those with CKD, and physical inactivity was associated with increased mortality [30]. Another observational study of NHANES data found that adults with CKD were sedentary nearly two-thirds of the time, and replacing sedentary time with an increase in light PA showed survival benefit [31]. Our results showed that a low and a moderate amount of MVPA was associated with a lower risk of death. A moderate amount of MVPA was associated with CVE. A high amount of MVPA was significantly associated with reduced risk of ESKD among 1324 adults with eGFR < 60 mL/min/1.73 m^2^. A possible explanation is that MVPA may have various beneficial pathways for mental, physical, metabolic, and inflammatory function across the amount of MVPA in CKD.

The American Medical Association and the American College of Sports Medicine established an initiative that exercise is medicine [15]. The WHO has reported that physical inactivity is a leading cause of death and recommends that individuals increase their PA levels, including planned exercise and other PA in transport, house chores, playing, and recreational activities [32]. The FITT principle is important when prescribing an adequate dose of PA for individual adults with CKD based on age, sex, comorbidities, and goals of health care. In our study, the association of MVPA and mortality was prominent in older and female CKD adults. Therefore, MVPA programs should not follow a one-size-fits-all approach. Instead, MVPA recommendations should individualize for maximum health efficiency with small effort. The risk reduction is prominent and accrues even at a low amount of MVPA.

This study has some limitations. First, causal relationships between MVPA and each clinical outcome are difficult to demonstrate due to the observational cohort study design. Reverse causality is possible for the associations. Second, although IPAQ is a widely used questionnaire, there is a lack of validation in CKD. The IPAQ has undergone validity assessment in young to middle-aged participants in the general population. Third, a single self-reported health questionnaire and IPAQ at the baseline were used in our assessments, which could have led to recall bias. MVPA often changes over time in response to individual depression, work, and lifestyle. Fourth, participation in MVPA was influenced by facility accessibility, weather, and other socioeconomic factors. Fifth, we measured the MVPA time of both leisure and occupational activities. Much existing evidence of health benefits relates to leisure-time PA [25,26], and heavy occupational PA might be harmful to health. However, a meta-analysis found that a high level of leisure-time PA and a moderate level of occupational PA had a beneficial effect on cardiovascular health [33]. Despite those limitations, our study has several strengths. We analyzed cross-sectional and longitudinal data to find the clinical implications of MVPA with multiple adjustment factors. The comprehensive health history and laboratory data of the KNOW-CKD cohort were used. We investigated the amount and frequency of MVPA by FITT principles for a median 5.9-year follow up.

## 5. Clinical Implication

CKD adults with MVPA showed various beneficial outcomes compared to those with no MVPA across the amount of MVPA. A low and a moderate amount of MVPA (0–14.9 MET-hours/week) was associated with a lower risk of all-cause death. A moderate amount of MVPA (7.5–14.9 MET-hours/week) was associated with a lower risk of CVE. A high amount of MVPA (15.0–29.9 MET-hours/week) was associated with a lower risk of ESKD in adults with eGFR < 60 mL/min/1.73 m^2^. Age and sex modified the relationship between the amount of MVPA and clinical outcomes.

## 6. Conclusions

Our findings showed that adults with MVPA are associated with reduced risk of poor clinical outcomes across the amount of MVPA in CKD. Future studies could focus on the benefits of PA on clinical outcomes using FITT principles and consider differences in subgroups in a large population with CKD. PA should be tailored for each patient in accordance with lifestyle, social factors, and personal goals of outcomes [34].

## Figures and Tables

**Figure 1 jcm-10-03365-f001:**
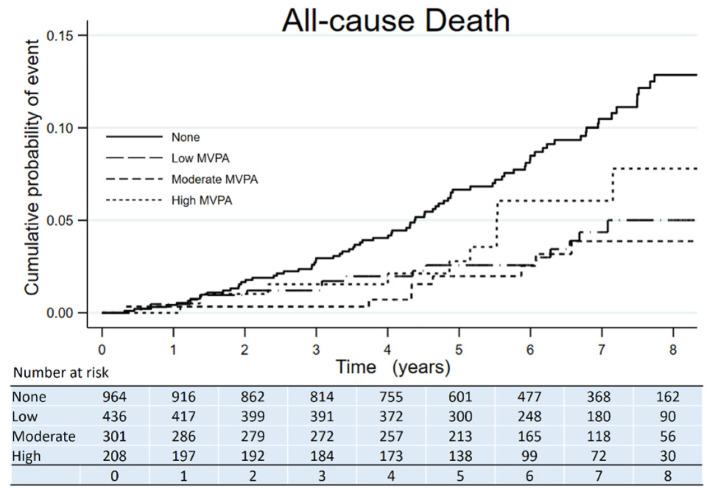
Kaplan–Meier survival curve of all-cause deaths across categories of MVPA amount in adults with nondialysis CKD.

**Figure 2 jcm-10-03365-f002:**
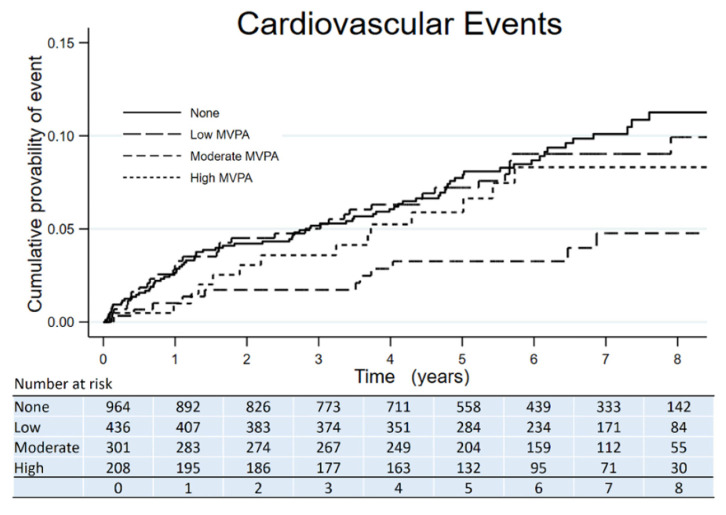
Kaplan–Meier survival curve of CVE across categories of MVPA amount in adults with nondialysis CKD.

**Figure 3 jcm-10-03365-f003:**
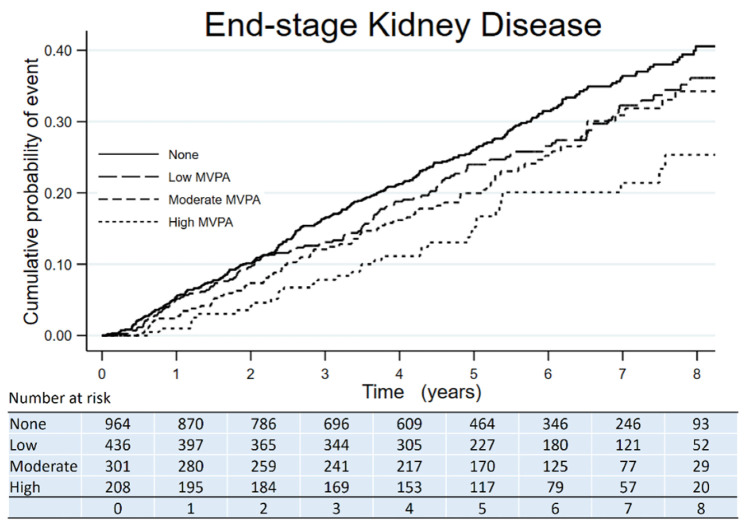
Kaplan–Meier survival curve of ESKD across categories of MVPA amount in adults with nondialysis CKD.

**Figure 4 jcm-10-03365-f004:**
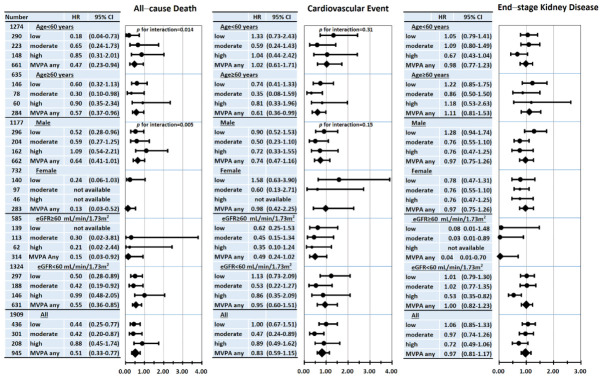
There were different risks of all-cause death and CVE across categories of MVPA amount according to subgroups in the multivariable-adjusted Cox proportional hazards model (Model 2).

**Table 1 jcm-10-03365-t001:** Baseline characteristics of the 1909 adults with nondialysis CKD according to the amount of MVPA.

	Amount of Moderate–Vigorous Physical Activity (MET-Hours/Week)				
	None	Low (<7.5)	Moderate (7.5–14.9)	High (15.0–29.9)	*p* Trend
Number, *n* (%)	964 (50.5)	436 (22.9)	301 (15.8)	208 (10.9)	
Age, years	54 ± ± 12	53 ± 13	52 ± 11	52 ± 12	0.003
Male sex, *n* (%)	515 (53.4)	296 (67.9)	204 (67.8)	162 (77.9)	<0.001
Current smoker, *n* (%)	152 (15.8)	72 (16.5)	39 (13.0)	45 (21.6)	0.17
Alcoholic consumption, *n* (%) (≥2 times/week)	116 (12.0)	61 (14.0)	54 (17.9)	44 (21.1)	0.001
Education					
≤High school graduate, *n* (%)	583 (60.5)	217 (49.8)	155 (51.5)	112 (53.8)	0.009
≥College graduate, *n* (%)	381 (39.5)	219 (50.2)	146 (48.5)	96 (45.2)	0.009
Work, employment, *n* (%)	502 (52.1)	265 (60.8)	202 (67.1)	144 (69.2)	<0.001
Cardiovascular disease, *n* (%)	167 (17.3)	60 (13.8)	38 (12.6)	27 (13.0)	0.031
Diabetes, *n* (%)	356 (36.9)	136 (31.2)	79 (26.3)	53 (25.5)	<0.001
Body mass index, kg/m^2^	24.4 ± 3.4	24.4 ± 3.4	24.7 ± 3.3	24.6 ± 3.1	0.25
eGFR, mL/min/1.73 m^2^	51 ± 32	55 ± 30	58 ± 32	54 ± 26	0.067
G1 ≥90	120 (12.5)	47 (10.8)	46 (15.3)	19 (9.1)	
G2 60–89	151 (15.7)	92 (21.1)	67 (22.3)	43 (20.7)	
G3 45–59	371 (38.5)	191 (43.8)	111 (36.9)	99 (47.6)	
G4 15–29	249 (25.8)	76 (17.4)	63 (20.9)	39 (18.8)	
G5 <15	73 (7.6)	30 (6.9)	14 (4.7)	8 (3.9)	
UACR, mg/g	389 (86–1176)	314 (77–1050)	320 (81–865)	382 (92–1042)	0.061
Systolic BP, mmHg	128 ± 17	127 ± 15	128 ± 15	129 ± 16	0.31
Diastolic BP, mmHg	77 ± 12	77 ± 10	78 ± 11	79 ± 10	0.023
Serum albumin, g/dL	4.16 ± 0.45	4.17 ± 0.44	4.23 ± 0.38	4.17 ± 0.36	0.18
CRP, mg/dL	0.65 (0.30–1.70)	0.63 (0.21–1.52)	0.61 (0.21–1.45)	0.71 (0.31–2.21)	0.88
Frequency of MVPA, per week	0	3 (2–5)	5 (3–7)	5 (2–7)	<0.001
Resistance exercise, per week	0	0 (0–2)	0 (0–3)	0 (0–3)	<0.001

Note: Values are the mean ± standard deviation or median (interquartile range). Abbreviations: MET, metabolic equivalent; eGFR, estimated glomerular filtration rate; UACR, urine albumin–creatinine ratio; BP, blood pressure; CRP, C-reactive protein.

**Table 2 jcm-10-03365-t002:** Association between demographic, social, and health factors in nondialysis CKD and MVPA.

	MVPA (>0 MET-Hours/Week)				MVPA (>7.5 MET-Hours/Week)			
	Crude OR (95% CI)	*p*	Multivariable OR (95% CI)	*p*	Crude OR (95% CI)	*p*	Multivariable OR (95% CI)	*p*
Sex, female	0.49 (0.41–0.59)	<0.001	0.46 (0.36–0.58)	<0.001	0.54 (0.43–0.67)	<0.001	0.50 (0.38–0.65)	<0.001
Age, years								
20–39	0.85 (0.62–1.45)	0.28	0.72 (0.52–0.99)	0.046	0.75 (0.53–1.05)		0.67 (0.47–0.96)	0.028
40–49	Reference		Reference		Reference		Reference	
50–59	0.83 (0.64–1.07)	0.15	0.84 (0.64–1.10)	0.23	0.88 (0.67–1.17)	0.095	0.91 (0.68–1.21)	0.53
60–75	0.66 (0.51–0.85)	0.001	0.71 (0.53–0.95)	0.022	0.60 (0.45–0.80)	0.012	0.66 (0.47–0.91)	0.012
Type 2 diabetes	0.70 (0.57–0.85)	<0.001	0.73 (0.59–0.91)	0.005	0.64 (0.51–0.81)	<0.001	0.67 (0.53–0.86)	0.002
Cardiovascular disease	0.73 (0.57–0.94)	0.013	0.74 (0.56–0.98)	0.032	0.75 (0.56–1.01)	0.065	0.83 (0.60–1.13)	0.24
eGFR, mL/min/1.73 m^2^								
G1 ≥90	0.70 (0.50–0.97)	0.034	0.83 (0.58–1.17)	0.28	0.85 (0.59–1.23)	0.41	1.04 (0.71–1.53)	0.85
G2 60–89	Reference		Reference		Reference		Reference	
G3 30–59	0.81 (0.63–1.04)	0.10	0.94 (0.72–1.22)	0.63	0.82 (0.62–1.08)	0.17	0.96 (0.72–1.29)	0.80
G4,5 <30	0.53 (0.41–0.70)	<0.001	0.69 (0.52–0.92)	0.013	0.64 (0.47–0.86)	0.004	0.83 (0.60–1.14)	0.26
Body mass index, kg/m^2^	1.01 (0.99–1.04)	0.38	1.00 (0.98–1.39)	0.57	1.02 (0.99–1.05)	0.17	1.02 (0.98–1.05)	0.20
Work, employed	1.68 (1.40–2.20)	<0.001	1.12 (0.97–1.11)	0.30	1.75 (1.41–2.16)	<0.001	1.19 (0.93–1.54)	0.15
Education, ≥College graduate	1.13 (1.07–1.20)	<0.001	1.04 (0.97–1.11)	0.26	1.06 (0.99–1.14)	0.081	0.90 (0.90–1.05)	0.52
Smoking, current	1.02 (0.80–1.30)	0.84	0.74 (0.57–1.01)	0.061	1.08 (0.82–1.42)	0.55	0.83 (0.63–1.11)	0.21

Abbreviations: MET, metabolic equivalent; OD, odds ratio; CI, confidence interval; eGFR, estimated glomerular filtration rate.

**Table 3 jcm-10-03365-t003:** Hazard ratios (HRs) for clinical outcomes according to the amount of MVPA.

Amount	Crude		Model 1		Model 2		Model 3	
	HR (95% CI)	*p*	HR (95% CI)	*p*	HR (95% CI)	*p*	HR (95% CI)	*p*
	**All-cause death**							
None	Reference		Reference		Reference		Reference	
Low	0.39 (0.23–0.67)	0.001	0.38 (0.22–0.67)	0.001	0.44 (0.25–0.77)	0.004	0.44 (0.25–0.78)	0.005
Moderate	0.30 (0.14–0.62)	0.001	0.36 (0.17–0.74)	0.006	0.41 (0.20–0.87)	0.019	0.42 (0.20–0.88)	0.022
High	0.56 (0.29–1.09)	0.087	0.73 (0.37–1.42)	0.35	0.88 (0.45–1.74)	0.71	0.90 (0.46–1.78)	0.77
Harrell’s C	0.609		0.808		0.844		0.845	
MVPA (>0)	0.40 (0.27–0.60)	0.001	0.44 (0.29–0.66)	0.001	0.51 (0.33–0.77)	0.001	0.51 (0.33–0.78)	0.002
	**Cardiovascular event**							
None	Reference		Reference		Reference		Reference	
Low	0.94 (0.64–1.40)	0.77	0.86 (0.57–1.30)	0.48	1.00 (0.67–1.51)	0.98	1.01 (0.68–1.55)	0.89
Moderate	0.41 (0.22–0.78)	0.007	0.44 (0.23–0.83)	0.011	0.47 (0.24–0.89)	0.020	0.47 (0.24–0.89)	0.021
High	0.79 (0.45–1.40)	0.43	0.72 (0.40–1.30)	0.28	0.89 (0.49–1.62)	0.70	0.89 (0.49–1.64)	0.72
Harrell’s C	0.554		0.713		0.771		0.774	
MVPA (>0)	0.72 (0.53–0.99)	0.044	0.69 (0.50–0.96)	0.028	0.81 (0.60–1.15)	0.23	0.82 (0.58–1.17)	0.27
	**End-stage kidney disease**							
None	Reference		Reference		Reference		Reference	
Low	0.84 (0.67–1.04)	0.10	0.94 (0.75–1.16)	0.55	1.06 (0.85–1.33)	0.58	1.05 (0.84–1.32)	0.65
Moderate	0.79 (0.62–1.02)	0.072	0.98 (0.75–1.26)	0.86	0.97 (0.74–1.26)	0.83	0.97 (0.74–1.26)	0.85
High	0.53 (0.38–0.75)	0.001	0.65 (0.45–0.95)	0.026	0.72 (0.49–1.06)	0.091	0.72 (0.49–1.06)	0.093
Harrell’s C	0.547		0.906		0.911		0.911	
MVPA (>0)	0.76 (0.63–0.90)	0.002	0.90 (0.75–1.07)	0.27	0.97 (0.81–1.17)	0.76	0.97 (0.80–1.16)	0.71

Model 1 was adjusted for age, sex, estimated glomerular filtration rate, and the natural log of albuminuria. Model 2 was adjusted for Model 1 variables plus education status, smoking, alcohol consumption, diabetes, cardiovascular disease, body mass index, systolic blood pressure, and serum albumin. Model 3 was adjusted for Model 2 variables plus statin use and angiotensin-converting enzyme inhibitor or angiotensin receptor blocker use.

## Data Availability

All data are available upon request.

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
