# Peer review of "Moderate–Vigorous Physical Activity and Clinical Outcomes in Adults with Nondialysis Chronic Kidney Disease"

_jcm, 2021, doi:10.3390/jcm10153365_

Round 1
Reviewer 1 Report
The health benefits of physical activity both in the general population and in chronic kidney disease are well known. The role of physical activity in CKD has been shown in several large studies including CRIC and NHANES. In this respect the manuscript provides only a limited number of new data and most the results are predictable. The manuscript brings the results from the subanalysis of the KoreaN cohort study for Outcome in patients With CKD (KNOW-CKD). The advantages of the study are the relatively large cohort and long-term follow-up, the disadvantages include an assessment of physical activity based on a a questionnaire instead of direct measurement methods that are widely available (e.g. using accelerometry). The authors should present the validation of the questionnaire in CKD population and the agreement of the results with the direct physical activity measurements. I could not find the information how many patients were in each stage of CKD. I would also like to see the detailed results of the analysis in a group of patients with advanced CKD and comparisons between less and more advanced CKD.
Author Response
We greatly appreciate the constructive comments of the reviewers, and we have carefully revised the manuscript. The revised sentences are highlighted in yellow.
Reviewer 1
-The authors should present the validation of the questionnaire in CKD population and the agreement of the results with the direct physical activity measurements.
We agree with your comment. That is the limitation of our study. A valid and reliable measure of PA is challenging. We used IPAQ which is inexpensive and easily adapted in large cohort study. Accelerometers are objective and reliable method compared with PA questionnaire. We have a plan to study PA using questionnaire and accelerometers on a small sample size of patients with CKD.
-I could not find the information how many patients were in each stage of CKD.
We add the numbers of adults according CKD GFR stages at baseline at Table 1 and 2.
-I would also like to see the detailed results of the analysis in a group of patients with advanced CKD and comparisons between less and more advanced CKD.
We add the clinical outcomes according to CKD stage (eGFR ≥60, <60 ml/min/1.73m2) at Figure 4. Subgroup analysis. Thank you for your suggestions and constructive comments. We have carefully revised the manuscript and hope you will find it acceptable for publication
Reviewer 2 Report
this is an observational study of almost 1900 CKD patients followed for 5.9 yrs with physical activity an demographics asssessed at baseline and associated with outcomes such as mortality, CVE and ESKD showing physical activity is associated with better outcome in this patient group
comments
please adjust the analyses for medication use (RAS blockade, antihypertensive medication statin use?)
for the subgroup analysis in fig 4 I would be concerned about the power of the analysis, could you at least provide information about number of subjects in each stratum (fex you have age ><60, but then 4 groups of PA?
as stated by authors reverse causality can not be excluded, please modify the conclusion accordingly
you could refer to the kidney benefits seen in lookahead although only in a diabetes population
Author Response
We greatly appreciate the constructive comments of the reviewers, and we have carefully revised the manuscript. The revised sentences are highlighted in yellow.
Reviewer 2
-please adjust the analyses for medication use (RAS blockade, antihypertensive medication statin use?)
We used the statin in 989 participants (52%) and used ARB or ACEI in 1638 participants (86%).
We added model 3 in Statistical Analyses.
àModel 3 was adjusted for model 2 variables plus statin use, and angiotensin-converting enzyme inhibitor or angiotensin receptor blocker use.
We analyze the Cox model proportional model (Model 3 added medication use) and added the HR and 95% CI in Table 3.
-for the subgroup analysis in fig 4 I would be concerned about the power of the Fanalysis, could you at least provide information about number of subjects in each stratum (fex you have age ><60, but then 4 groups of PA?
We agree with your comment. We tried to present the various association between MVPA and clinical outcomes in CKD subgroups. The sample sizes in several part of subgroup analysis are small and the statistic power may be low. We add the sample size in Figure 4 left part. The groups of small size (<100 participants: Age <60 moderate, high, Female moderate high, eGFR>60 high) mark HR with small dots.
-as stated by authors reverse causality can not be excluded, please modify the conclusion accordingly
We remove and exchange the sentences in Conclusion
Our findings reinforce the health benefits of MVPA in adults with CKD. We showed that MVPA is associated with the reducing risk of poor clinical outcomes across the amount of MVPA.
à Our findings showed that MVPA is associated with the lower risk of poor clinical outcomes across the amount of MVPA.
you could refer to the kidney benefits seen in lookahead although only in a diabetes population
Thank you for your comment. According to our results, we think the kidney benefit by MVPA is less than mortality or cardiovascular benefit by MVPA. Thank you for your suggestions and constructive comments. We have carefully revised the manuscript and hope you will find it acceptable for publication
Round 2
Reviewer 1 Report
The issue of the lack of validation of the IPAQ questionnaire that was used in the study needs to be commented in detail in the Discussion. The points to discuss include a lack of IPAQ validation for CKD (advanced stages of the disease) and for subjects older that 69 that were included in the study.
Author Response
The issue of the lack of validation of the IPAQ questionnaire that was used in the study needs to be commented in detail in the Discussion. The points to discuss include a lack of IPAQ validation for CKD (advanced stages of the disease) and for subjects older that 69 that were included in the study.
We are sorry for our unclearness and limited knowledge about the IPAQ validity.
We add the sentence to the limitations of the discussion part. (6 page, line 139-141)
Second, although IPAQ is widely used questionnaire, it is a lack of validation in CKD. The IPAQ has undergone validity assessment in young to middle aged in the general population.
Thank you for your kind comment. We have revised the manuscript and we sincerely hope the manuscript is now acceptable for publication